# The Validity of Benchmark Dose Limit Analysis for Estimating Permissible Accumulation of Cadmium

**DOI:** 10.3390/ijerph192315697

**Published:** 2022-11-25

**Authors:** Soisungwan Satarug, David A. Vesey, Glenda C. Gobe, Aleksandra Buha Đorđević

**Affiliations:** 1Kidney Disease Research Collaborative, Translational Research Institute, Brisbane 4102, Australia; 2Department of Nephrology, Princess Alexandra Hospital, Brisbane 4102, Australia; 3School of Biomedical Sciences, The University of Queensland, Brisbane 4072, Australia; 4NHMRC Centre of Research Excellence for CKD QLD, UQ Health Sciences, Royal Brisbane and Women’s Hospital, Brisbane 4029, Australia; 5Department of Toxicology “Akademik Danilo Soldatović”, Faculty of Pharmacy, University of Belgrade, 11000 Belgrade, Serbia

**Keywords:** benchmark dose, BMD lower confidence limit, BMDL, BMD upper confidence limit, BMDU, cadmium, β_2_-microglobulin, eGFR, N-acetyl-β-D-glucosaminidase, NOAEL

## Abstract

Cadmium (Cd) is a toxic metal pollutant that accumulates, especially in the proximal tubular epithelial cells of kidneys, where it causes tubular cell injury, cell death and a reduction in glomerular filtration rate (GFR). Diet is the main Cd exposure source in non-occupationally exposed and non-smoking populations. The present study aimed to evaluate the reliability of a tolerable Cd intake of 0.83 μg/kg body weight/day, and its corresponding toxicity threshold level of 5.24 μg/g creatinine. The PROAST software was used to calculate the lower 95% confidence bound of the benchmark dose (BMDL) values of Cd excretion (E_Cd_) associated with injury to kidney tubular cells, a defective tubular reabsorption of filtered proteins, and a reduction in the estimated GFR (eGFR). Data were from 289 males and 445 females, mean age of 48.1 years of which 42.8% were smokers, while 31.7% had hypertension, and 9% had chronic kidney disease (CKD). The BMDL value of E_Cd_ associated with kidney tubular cell injury was 0.67 ng/L of filtrate in both men and women. Therefore, an environmental Cd exposure producing E_Cd_ of 0.67 ng/L filtrate could be considered as Cd accumulation levels below which renal effects are likely to be negligible. A reduction in eGFR and CKD may follow when E_Cd_ rises from 0.67 to 1 ng/L of filtrate. These adverse health effects occur at the body burdens lower than those associated with E_Cd_ of 5.24 µg/g creatinine, thereby arguing that current health-guiding values do not provide a sufficient health protection.

## 1. Introduction

Cadmium (Cd) is an environmental toxicant of worldwide public health significance because numerous population-based studies suggest that exposure to Cd even at low levels adversely impacts the functions of most organs of the body [1]. Volcanic emissions, biomass and fossil fuel combustion, and cigarette smoke are sources of environmental Cd pollution [2,3,4,5,6]. Cd in cigarette smoke as a volatile metallic form and oxide (CdO) has a particularly high transmission rate [7]. The utility of Cd in many industrial processes, and the use of phosphate fertilizers contaminated with Cd cause also a widespread dispersion of this toxic metal in the environment and subsequently the food chains [8,9,10,11,12].

Foods that are frequently consumed in large quantities such as rice, potatoes, wheat, leafy salad vegetables, and other cereal crops form the most significant dietary sources of Cd [13,14,15]. Seafood (shellfish), molluscs and crustaceans are additional dietary Cd sources [16,17].

Cd accumulates most extensively in the proximal tubular epithelial cells of the kidneys, and the kidney burden of Cd as µg/ g tissue weight increases with age [18,19]. Approximately 0.001–0.005% of Cd in the body is excreted in the urine each day, and the biological half-life of Cd in the kidney cortex is estimated to be 30 years for non-smokers [19,20]. In Japanese residents of a Cd pollution area, the average half-life of the metal among those with a lower body burden (urinary Cd < 5 µg/L) was 23.4 years; in those with a higher body burden (urinary Cd > 5 µg/L), the average half-life was 12.4 years [21,22]. Thus, the lower the body burden, the longer the half-life of Cd.

To safeguard against excessive dietary Cd exposure, health guidance values have been established by the Joint FAO/WHO Expert Committee on Food Additives and Contaminants (JECFA) [23]. Based on a risk assessment model that assumed an increase in the excretion of β_2_-microglobulin (β_2_M, E_β2M_) above 300 μg/g creatinine to be the “critical” toxicity endpoint, the tolerable intake level of Cd was set at 0.83 µg/kg body weight/day, and a urinary Cd excretion of 5.24 µg/g creatinine was considered to be the toxicity threshold level [23]. However, E_β2M_ levels of 100–299, 300–999, and ≥1000 μg/g creatinine were found to be associated with 4.7-, 6.2- and 10.5-fold increases in the risk of an estimated GFR (eGFR) ≤ 60 mL/min/1.73 m^2^, commensurate with chronic kidney disease (CKD) [24]. Thus, E_β2M_ above 300 μg/g creatinine does not appear to be an early warning sign of the nephrotoxicity of Cd. The utility of β_2_M excretion as a toxicity criterion to derive Cd health guidance values is questionable.

Presently, the benchmark dose (BMD) has often been used to derive health guidance values instead of the no-observed-adverse-effect level (NOAEL) [25,26,27]. The BMD is a dose level, derived from an estimated dose–response curve, associated with a specified change in response, termed benchmark response (BMR) which can be set at 1%, 5%, and 10% as required. BMD rectifies some of the shortcomings of the NOAEL [28,29]. For a continuous endpoint, the lower 95% confidence bound of BMD, termed BMDL value derived at 5% BMR has been viewed as NOAEL equivalent or the level below which adverse health effect could be negligible [25].

The present study had three aims. The first aim was to evaluate the validity of current kidney toxicity threshold level of E_Cd_ at 5.24 μg/g creatinine. The second aim was to define the BMDL values (NOAEL equivalents) of E_Cd_ from three indicators of renal effects; injury to kidney tubular epithelial cells, a defective tubular reabsorption of β_2_M, and eGFR decline. Kidney tubular cell injury was indicated by an increase in excretion of N-acetyl-β-D-glucosaminidase (NAG, E_NAG_). We postulated that the BMDL value of E_Cd_ for the E_NAG_ endpoint was the lowest, given that the E_NAG_ represents an early sign of the cytotoxicity of Cd. The third aim was to demonstrate an impact of methods used to normalize Cd, NAG and β_2_M excretion rates. Accordingly, we assessed the validity of BMDL values of E_Cd_ obtained from normalizing excretion rates of Cd, NAG and β_2_M to creatinine clearance (C_cr_) as E_x_/C_cr_ and to the excretion of creatinine (E_cr_) as E_x_/E_cr_, where x = Cd, NAG or β_2_M.

## 2. Materials and Methods

### 2.1. Study Subjects

We assembled archived data from 289 men and 445 women who were drawn from Bangkok (a low-exposure area) and from Cd contaminated areas of Mae Sot District (a high-exposure area) in Thailand [30]. The Ethical Committee of Chulalongkorn University and the Mae Sot Hospital Ethical Committee approved the study protocol. At the time of recruitment, all participants had lived at their current addresses for at least 30 years, and all gave informed consent prior to participation. Exclusion criteria were pregnancy, breastfeeding, a history of metalwork, and a hospital record or physician’s diagnosis of an advanced chronic disease. Smoking, diabetes, hypertension, regular use of medications, educational level, occupational and family health history were ascertained by a questionnaire. Diabetes was defined as fasting plasma glucose levels ≥ 126 mg/dL or a physician’s prescription of anti-diabetic medications. Hypertension was defined as systolic blood pressure ≥ 140 mmHg, diastolic blood pressure ≥ 90 mmHg, and/or a physician’s diagnosis and prescription of anti-hypertensive medications.

### 2.2. Blood and Urine Sampling and Analyses

Simultaneous blood and urine sampling is required to normalize E_Cd_, E_β2M_ and E_NAG_ to C_cr_ Accordingly, second morning urine samples were collected after an overnight fast, and whole blood samples were obtained within 3 h after the urine sampling. Aliquots of urine, whole blood and plasma were stored at −20 °C or −80 °C for later analysis. The assay for urine and plasma concentrations of creatinine ([cr]_u_ and [cr]_p_) was based on the Jaffe reaction. The urinary NAG assay was based on colorimetry (NAG test kit, Shionogi Pharmaceuticals, Sapporo, Japan). The urinary β_2_M assay was based on the latex immunoagglutination method (LX test, Eiken 2MGII; Eiken and Shionogi Co., Tokyo, Japan).

For the Bangkok group, [Cd]_u_ was determined by inductively-coupled plasma mass spectrometry (ICP/MS, Agilent 7500, Agilent Technologies, Santa Clara, CA, USA) because it had the high sensitivity required to measure very low Cd concentrations. Multi-element standards (EM Science, EM Industries, Inc., Newark, NJ, USA) were used to calibrate the Cd analyses. The accuracy and precision of those analyses were ascertained with reference urine (Lyphochek^®^, Bio-Rad, Sydney, Australia). The low limit of detection (LOD) of urine Cd, calculated as 3 times the standard deviation of blank measurements was 0.05 µg/L. The Cd concentration assigned to samples with Cd below the detection limit was the detection limit divided by the square root of 2 [31].

Based on E_Cd_ data reported for Mae Sot residents that could be reliably quantified by atomic absorption spectrophotometry (AAS), the [Cd]_u_ in samples from Mae Sot group was determined by AAS (Shimadzu Model AA-6300, Kyoto, Japan). Urine standard reference material No. 2670 (National Institute of Standards, Washington, DC, USA) was used for quality assurance and control purposes. The LOD of Cd quantitation, calculated as 3 times the standard deviation of blank measurements was 0.06 µg/L. None of the urine samples from this group was found to have a [Cd]_u_ below the detection limit.

### 2.3. Estimated Glomerular Filtration Rate (eGFR)

The GFR is the product of nephron number and mean single nephron GFR, and in theory the GFR is indicative of nephron function [32,33,34]. In practice, the GFR is estimated from established chronic kidney disease-epidemiology collaboration (CKD-EPI) equations, and is reported as eGFR.

Male eGFR = 141 × [serum creatinine/0.9]^y^ × 0.993^age^, where y = −0.411 if serum creatinine ≤ 0.9 mg/dL or −1.209 if serum creatinine > 0.9 mg/dL.

Female eGFR = 144 × [serum creatinine/0.7]^y^ × 0.993^age^, where y = −0.329 if serum creatinine ≤ 0.7 mg/dL or −1.209 if serum creatinine > 0.7 mg/dL.

These CKD-EPI equations have been validated with inulin clearance, and they are considered to be the most accurate eGFR calculating equations [35].

### 2.4. Normalization of E_Cd_, E_β2M_ and E_NAG_ to E_cr_ and C_cr_

E_x_ was normalized to E_cr_ as [x]_u_/[cr]_u_, where x = Cd, β_2_M or NAG; [x]_u_ = urine concentration of x (mass/volume); and [cr]_u_ = urine creatinine concentration (mg/dL). The ratio [x]_u_/[cr]_u_ was expressed in μg/g of creatinine.

E_x_ was normalized to C_cr_ as E_x_/C_cr_ = [x]_u_[cr]_p_/[cr]_u_, where x = Cd, β_2_M or NAG; [x]_u_ = urine concentration of x (mass/volume); [cr]_p_ = plasma creatinine concentration (mg/dL); and [cr]_u_ = urine creatinine concentration (mg/dL). E_x_/C_cr_ was expressed as the excretion of x per volume of filtrate [36]. Derivation of an equation for C_cr_ normalization is in Appendix B. Demonstration that E_Cd_/C_cr_ is unaffected by muscle mass is in Appendix C.

### 2.5. Benchmark Dose Computation and Benchmark Response (BMR) Setting

We used the web-based PROAST software version 70.1 (https://proastweb.rivm.nl (accessed on 9 June 2022) to compute the BMD figures for Cd exposure as E_Cd_/E_cr_ or E_Cd_/C_cr_ associated with three effect markers, eGFR, E_NAG,_ and E_β2M_. A specific effect size, termed BMR, was set at 5% for all continuous endpoints, given that the BMDL values obtained could serve as the point of departure to derive health guidance values [25,26,27]. BMD values were computed from fitting datasets to multiple dose–response models such as inverse exponential, natural logarithmic, exponential, and Hill models. BMDL/BMDU values were computed from those showing a statistically significant dose–response relationship. Model averaging was used instead of a single model analysis to account for model uncertainty.

The BMR was set at a 10% increase in the prevalence of the following quantal endpoints; eGFR ≤ 60 mL/min/1.73 m^2^, E_NAG_/E_cr_ ≥ 4 U/g creatinine, E_β2MG_/E_cr_ ≥ 300 µg/g creatinine, E_NAG_/C_cr_ × 100 ≥ 4 U/L filtrate, and E_β2MG_/C_cr_ × 100 ≥ 300 µg/L filtrate. These endpoints were indicative of the presence of CKD, toxic injury to kidney tubular epithelial cells, and a reduction in tubular reabsorption of filtered protein, respectively. BMD values were calculated from fitting datasets to multiple dose–response models that included two-stage, logarithmic logistic, Weibull, logarithmic probability, gamma, exponential and Hill models.

The BMDL and BMDU corresponded to the lower bound and upper bound of the 95% confidence interval (CI) of BMD. The BMDL/BMDU were from model averaging using bootstrap with 200 repeats. A wider BMDL-BMDU difference, a higher statistical uncertainty in the dataset [28,29,37].

### 2.6. Statistical Analysis

Data were analyzed with IBM SPSS Statistics 21 (IBM Inc., New York, NY, USA). The distributions of eGFR and the excretion of Cd, β_2_M and NAG were examined for skewness, and those showing rightward skewing were subjected to logarithmic transformation before analysis. The departure of a given variable from a normal distribution was assessed with the one-sample Kolmogorov–Smirnov test. The Kruskal–Wallis test was used to compare mean differences across three eGFR groups. The Mann–Whitney U-test was used to compare mean differences between two groups. The Chi-square test was used to determine differences in percentage and prevalence data. For each analysis, *p*-values ≤ 0.05 for two-tailed tests were assumed to indicate statistical significance.

## 3. Results

### 3.1. Characteristics of Males and Females of Low- and High-Exposure Locations

Among 734 participants, 100 males and 100 females were residents of a low-exposure area (Bangkok), and 189 males and 345 females were residents of a high-exposure area (Mae Sot) residents (Table 1).

The overall mean age of 734 participants was 48.1 years. Of the total participants, 42.8% were smokers to include those who had quite less than 10 years. Another 31.7% of participants had hypertension and only 1.5% had diabetes. The overall % of CKD was 9%. None of subjects of the low-exposure group had CKD, while the % of CKD among males and females of the high-exposure group were 15.3% and 10.7%, respectively.

The overall % of subjects with a sign of tubular cell injury evident from E_NAG_/E_cr_ ≥ 4 U/g creatinine was 76.2%, while the overall % of subjects showing a sign of tubular dysfunction was 39.8%.

Smoking was particularly high among males in the Mae Sot group (81.5%) than in males of the Bangkok group (47%). All females of the Bangkok group were non-smokers, while 32.8% of female residents of Mae Sot smoked cigarettes.

Mean E_Cd_ for smokers in Mae Sot was 18.3-fold higher than that of smokers in Bangkok (5.87 vs. 0.32 µg/L, *p* < 0.001). The mean E_Cd_ was 24.9-fold higher in non-smokers of the Mae Sot than non-smokers of Bangkok (5.22 vs. 0.21 µg/L, *p* < 0.001).

For the Bangkok group, mean [cr]_p_ was lower in females than males (*p* < 0.001) as was mean E_NAG_/C_cr_ (*p* = 0.023). For the Mae Sot group, mean [cr]_p_ and mean [cr]_u_ in females were lower than males (*p* < 0.001 for both parameters) as were mean [Cd]_u_ (*p* = 0.020) and mean E_Cd_/C_cr_ (*p* = 0.021). In contrast, mean E_NAG_/E_cr_ in females was higher than that of males (*p* < 0.001).

### 3.2. Characteristics of Study Subjects, Stratified by eGFR Values

In current practice, staging of CKD is based on eGFR values where CKD stages 1, 2, 3a, 3b, 4, and 5 correspond to eGFR of 90–119, 60–89, 45–59, 30–44, 15–29, and <15 mL/min/1.73 m^2^, respectively [33]. For dichotomous comparisons, CKD is defined as eGFR ≤ 60 mL/min/1.73 m^2^ [33].

Table 2 provides data from 734 participants stratified by eGFR levels 1, 2 and 3 corresponding to eGFR ≥ 90, 61–89, and ≤ 60 mL/min/1.73 m^2^, respectively.

The percentages of females and those with hypertension in each eGFR group did not differ, while % of diabetes (6.1%) and smoking (63.6%) were the highest in the lowest eGFR group. None of subjects from a low exposure area had low eGFR. Subjects of the low eGFR group was the oldest (mean age 63.2), who also had the lowest mean BMI [(*p* = 0.020). Mean [cr]_p_, mean [cr]_u_ and mean [Cd]_u_ all were the highest in the eGFR level 3, middle in eGFR level 2, and the lowest in eGFR level 3 as were the means for E_cr_- and C_cr_-normalized E_Cd_, E_β2MG_, E_NAG_.

### 3.3. BMDL and BMDU Values of Cadmium Excretion from E_cr_-Normalized Dataset

The BMDL and BMDU values of E_Cd_/E_cr_ computed from three kidney toxicity endpoints; tubular cell injury, tubular dysfunction and eGFR reduction are provided separately for males and females (Table 3).

In continuous dose–response analyses, the respective BMDL/BMDU of E_Cd_/E_cr_ associated with a tubular cell injury in men and women were 0.060/0.504 and 0.069/0.537 µg/g creatinine (Appendix A), while the corresponding BMDL/BMDU of E_Cd_/E_cr_ associated with a tubular dysfunction were 0.019/0.411 and 0.022/0.491 µg/g creatinine (Appendix A). The BMDL (NOAEL equivalent) values of E_Cd_/E_cr_ associated with these tubular toxicity indicators could be reliably determined, given that the BMDU/BMDL ratios ranged between 8.44 and 23. In contrast, the BMDU/BMDL ratios of E_Cd_/E_cr_ associated with a 5% decrease in eGFR were extremely large (Appendix A). Thus, the respective BMDL/BMDU values of E_Cd_/E_cr_ associated with eGFR effect in men and women of 1.57 × 10^−6^/0.427 and 1.88 × 10^−5^/0.495 µg/g creatinine were not reliably determined.

In quantal dose–response analyses (Appendix A), the respective BMDU/BMDL ratios of E_Cd_/E_cr_ associated with a 10% increase in the prevalence of tubular cell injury in men and women were 1.48 × 10^−3^/0.56 and 3.17 × 10^−4^/0.706 µg/g creatinine. These BMDL values were not reliable due to the extremely large BMDL/BMDU ratios of E_Cd_/E_cr_ in both men and women. The respective BMDL/BMDU values of E_Cd_/E_cr_ associated with a 10% incidence of an increase in prevalence of tubular dysfunction were reliably determined as 0.469/0.973 and 0.733/1.29 µg/g creatinine. Likewise, reliable BMDL/BMDU values of E_Cd_/E_cr_ associated with a 10% increase in the prevalence of CKD in men and women were 3.26/7.46 and 4.98/9.68 µg/g creatinine.

### 3.4. BMDL and BMDU Valuse of Cadmium Excretion from C_cr_-Normalized Dataset

The BMDL and BMDU values of E_Cd_/C_cr_ computed from three kidney toxicity endpoints; tubular cell injury, tubular dysfunction and eGFR reduction are provided separately for males and females (Table 4).

In continuous dose–response analyses, the respective BMDL/BMDU of E_Cd_/C_cr_ × 100 associated with a tubular cell injury in men and women were 0.067/0.394, and 0.067/0.399 µg/L filtrate (Appendix A), while the corresponding BMDL/BMDU of E_Cd_/C_cr_ × 100 associated with a 5% reduction in reabsorption of β_2_M were 0.41 × 10^−4^/0.038, and 0.16 × 10^−3^/0.040 µg/L (Appendix A). The BMDL (NOAEL equivalent) values of E_Cd_/C_cr_ × 100 associated with a defective tubular reabsorption in men and women could not be reliably determined, due extremely large BMDU/BMDL ratios, indicative of a high degree of statistical uncertainty. In contrast, the BMDU/BMDL values of E_Cd_/C_cr_ × 100 associated with a 5% eGFR decline were determined with a sufficiently high degree of statistical certainty as 0.080/0.714 and 0.098/0.812 µg/L filtrate in men and women, respectively (Appendix A).

In quantal dose–response analyses (Appendix A), the respective BMDU/BMDL ratios of E_Cd_/C_cr_ associated with a 10% increase in the prevalence of tubular cell injury in men and women 2.54/4.58, and 3.66/6.14 µg/L filtrate, respectively. The respective BMDL/BMDU values of E_Cd_/C_cr_ × 100 associated with a 10% incidence of an increase in prevalence of defective tubular dysfunction were 0.456/0.807 and 0.5/0.805 µg/ L filtrate, respectively. Likewise, reliable BMDL/BMDU values of E_Cd_/C_cr_ × 100 associated with a 10% increase in the prevalence of CKD in men and women were 5.06/8.66 µg/ L filtrate in men and 5.15/7.62 µg/L filtrate in women.

## 4. Discussion

To the best of our knowledge, we herein provide, for the first time, BMDL values of Cd excretion levels that are derived simultaneously from three toxicity endpoints; injury to kidney tubular epithelial cells, a defective tubular reabsorption of β_2_MG, and eGFR decline. As age distribution histogram indicates (Appendix A), these BMDL values of permissible Cd accumulation levels can be generalized to an entire adult population. A Cd accumulation level producing E_Cd_/C_cr_ of 0.67 ng/L filtrate in men and women could be considered as Cd accumulation levels below which renal effects are likely to be negligible (Appendix A). A reduction in eGFR may follow when Cd accumulation levels rise to 0.80 ng/L filtrate in men and 0.98 ng/L filtrate in women (Appendix A).

The reported BMD values of E_Cd_/E_cr_ associated with adverse effects on kidneys vary among studies. The variability was most likely due to differences in age of subjects included in an analysis as well as differences in the dose–response models applied to the datasets [25,26,27]. It is noteworthy that most BMD values of E_Cd_ associated with adverse kidney effects were derived from those age 50 years or older. Consequently, most BMD figures previously published were not generalizable to the entire adult population. For a quantal endpoint, different cut-off values constituted an additional source of variable BMD values. There is a need to standardize the BMD approach to advance Cd toxicological risk assessment. Age should form an important consideration in deriving BMDL.

In the present study, we used E_NAG_/E_cr_ of 4 U/g creatinine as the cut-off value to reflect injury to kidney tubular epithelial cells (Appendix A). E_NAG_ is considered to be proportional to the number of surviving nephrons because NAG in urine originated from injured or dying tubular cells [38,39,40]. In a United Kingdom (U.K.) study, a dose–response relationship was observed between E_Cd_ and E_NAG_; E_Cd_/E_cr_ of 0.5 μg/g creatinine was associated with 2.6- and 3.6-fold increases in the prevalence of E_NAG_/E_cr_ > 2 U/g creatinine, as compared with a E_Cd_/E_cr_ of 0.3 and < 0.5 μg/g creatinine, respectively [41]. The BMDL/BMDU figures derived by us from E_NAG_ endpoint were in line with the U.K. study. Apparently, these NOAEL equivalents of E_Cd_/E_cr_ associated with tubular epithelial cell injury were lower than E_Cd_/E_cr_ of 5.24 μg/g creatinine, suggested to be a nephrotoxicity threshold level of Cd [18].

The BMDL values of E_Cd_/E_cr_ derived from kidney tubular cell injury endpoint were 0.060 and 0.069 µg/g creatinine in men and women, respectively (Appendix A). The respective BMDL values of E_Cd_/E_cr_ associated with a 5% decrease of tubular reabsorption of β_2_M in men and women are 0.019 and 0.022 µg/g creatinine (Appendix A). In contrast, the BMDL values of E_Cd_/E_cr_ associated with a 5% eGFR decline could not be reliably determined (U/L ratios of 10^4^–10^5^). These findings were consistent with data from a conventional dose–response analysis, where an inverse relationship between E_Cd_ and eGFR became statistically significant only when E_Cd_ was normalized to C_cr_ [42].

Because of high variance in datasets introduced by E_cr_-normalization, the effect of Cd on eGFR was not realized. In a systematic review and meta-analysis of pooled data from 28 studies, the risk of proteinuria was increased by 1.35-fold when comparing the highest vs. lowest category of Cd dose metrics, but an increase in the risk of low eGFR was statistically insignificant (*p* = 0.10) [43]. An erroneous conclusion that chronic Cd exposure was not associated with a progressive eGFR reduction was also made in another systematic review [44].

In a meta-analysis of pooled data from 30 publications, the BMD of E_Cd_/E_cr_ for tubular injury (E_NAG_) was 1.67 μg/g creatinine [45]. In another study using data from 469 men and 465 women who were resident of Zhejiang Province, China, the E_Cd_/E_cr_ BMD/BMDL figures based on E_β2M_ and E_NAG_ were 1.24/0.62 and 0.85/0.49 μg/g creatinine in men, respectively [46]. The corresponding figures for women were 1.35/0.64 and 1.36/0.65 μg/g creatinine, respectively. Of note, in men only, E_Cd_/E_cr_ BMD/BMDL figures for E_β2MG_ were higher than those for E_NAG_. This was seen in a Swedish study [47] and our datasets.

In a study of data from 410 men and 418 women, aged 40–59 years, who were residents of non-pollution areas of Japan, the E_Cd_/E_cr_ BMD figures for E_β2M_ endpoint were 0.6–1.2 and 0.6–2.3 µg g creatinine in men and women, respectively [48].

According to a quantal endpoint of E_β2M_/E_cr_ ≥ 300 μg/g creatinine, the BMDL value E_Cd_/E_cr_ values were 0.469 and 0.733 µg/g creatinine in men and women, respectively (Appendix A). These data imply that the prevalence of Cd-induced tubular dysfunction was likely to be smaller than 10% at E_Cd_/E_cr_ 0.469 µg/g creatinine in men, and 0.733 µg/g creatinine in women. Of concern, these E_Cd_/E_cr_ levels have been associated with increased risk of CKD in the general populations in the U.S. [49] Spain [50], Taiwan [51] and China [52,53]. Arguably, E_β2M_ is not a sensitive toxicity endpoint and consequently the dietary exposure guideline of tolerable intake at 0.83 µg/kg body weight/day derived from E_β2M_/E_cr_ ≥ 300 µg/g creatinine endpoint does not afford sufficient health protection.

The BMD values for E_NAG_ and eGFR reduction in a study of 790 Swedish women, aged 53–64 years, were 0.5–0.8 and 0.7–1.2 μg/g creatinine, respectively [47]. A Cd-induced kidney tubular cell injury and eGFR reduction could be more suitable endpoints for health risk calculation than E_β2M_. In clinical trials, the successful treatment of kidney disease is judged by the attenuation of a decline in eGFR [33,54]. CKD is a progressive syndrome with high morbidity and mortality, and affects 8% to 16% of the world’s population. The continuing rise in the incidence of CKD globally is a cause of concern [54].

It has long been viewed that excreted Cd included Cd molecules that pass through the glomerular filtration membrane into the filtrate but are not reabsorbed. However, it is noteworthy that the excreted Cd originates from injured or dying tubular cells [40]. In an analogy to E_NAG_, Cd excretion is a manifestation of the cytotoxicity of its accumulation in kidneys’ tubular cells. In a histopathological examination of kidney biopsies from healthy kidney transplant donors [55], the degree of tubular atrophy was positively associated with the level of Cd accumulation. Tubular atrophy was observed at relatively low Cd levels [55].

The reductions in eGFR due to Cd nephropathy have often been attributed to glomerular injury. However, current evidence suggests that sufficient tubular injury disables glomerular filtration and leads to nephron atrophy and GFR loss [40,56]. Cd inflicts tubular cell injury at low intracellular concentrations, and the toxicity intensifies as Cd concentration rises [40]. The Cd-induced tubular injury disables glomerular filtration, leading to nephron atrophy, glomerulosclerosis, and interstitial inflammation and fibrosis. A reduction in tubular reabsorption of filtered proteins, retinal binding protein (RBP) and β_2_MG follows tubular atrophy and nephron loss. An increase of β_2_M excretion to levels above 300 µg/g creatinine is speculatively due to effects of Cd on both tubular reabsorption and nephron number.

Figure 1 depicts the pathogenesis of Cd-induced nephropathy that originates from tubular injury deduced from the BMDL values of E_Cd_/C_cr_ provided in Table 4.

In theory, an acceptable level of environmental exposure to Cd should be derived from the most sensitive toxicity endpoint, which is the one with the lowest BMDL value [27]. In the present study, BMDL for a 5% increase of E_β2M_/C_cr_ could not be reliably determined (Table 4), thereby arguing that the utility of E_β2M_ as a toxicity criterion is not appropriate. The BMDL value of E_Cd_ for E_NAG_ endpoint appeared to be the lowest (Table 4). Thus, a tolerable intake level of Cd derived from E_NAG_/C_cr_ endpoint will be sufficiently low to protect against Cd-induced toxic tubular cell injury. This E_NAG_-based tolerable intake level of Cd will be lower than a tolerable intake level of 0.83 µg/kg body weight per day that was derived from E_β2M_ endpoint.

The protein β_2_M, with a molecular weight of 11,800 Da, is synthesized and shed by all nucleated cells in the body [57]. Owing to its small mass, β_2_M is filtered freely by the glomeruli and is reabsorbed almost completely by the kidney’s tubular epithelial cells [58,59,60]. Increased β_2_M excretion has been used as an indicator of impaired tubular re-absorptive function. It is noteworthy that β_2_M production rises in response to many inflammatory and neoplastic conditions [61] and this compromises its utility as a marker of tubulopathy. If reabsorption rates of β_2_M per nephron remain constant as its production rates change, excretion will vary directly with its production [Appendix D]. If the production and reabsorption per nephron remain constant as nephrons are lost, the excretion of β_2_M will rise [Appendix E].

Consistent with the above notions are data in Table 2 where mean E_β2M_/C_cr_ in those with low eGFR was 90-fold higher than those with eGFR ≥ 90 mL/min/1.73 m^2^, while there was only a 3-fold increase in E_β2M_/C_cr_ in those with moderate eGFR (eGFR 61–89 mL/min/1.73 m^2^). E_β2M_ is a function of nephron numbers and tubular reabsorption activity, a substantial increase in E_β2M_ can be expected when nephrons are lost [24]. Accordingly, the sue of E_β2M_ as a “critical” effect of Cd accumulation in kidney tubular cells is inappropriate.

## 5. Conclusions

The BMDL/BMDU values of E_Cd_/C_cr_ associated with a 5% increase of kidney tubular cell injury are 0.67/3.94 and 0.67/3.99 ng/L filtrate in men and women, respectively. The respective BMDL/BMDU values of E_Cd_/C_cr_ associated with a 5% eGFR decline in men and women are 0.80/7.14 and 0.98/8.12 ng/L filtrate. A Cd accumulation level resulting in Cd excretion at 0.67 ng/L filtrate in men and women could be viewed as NOAEL equivalent and is a maximally permissible Cd accumulation level. A reduction in eGFR and CKD may follow when Cd excretion levels exceed this maximal permissible accumulation level of 0.67 µg/L of filtrate. Health guidance values for Cd derived from these NOAEL equivalents of Cd excretion levels are more likely to be low enough to carry a negligible adverse effect on kidneys.

## Figures and Tables

**Figure 1 ijerph-19-15697-f001:**
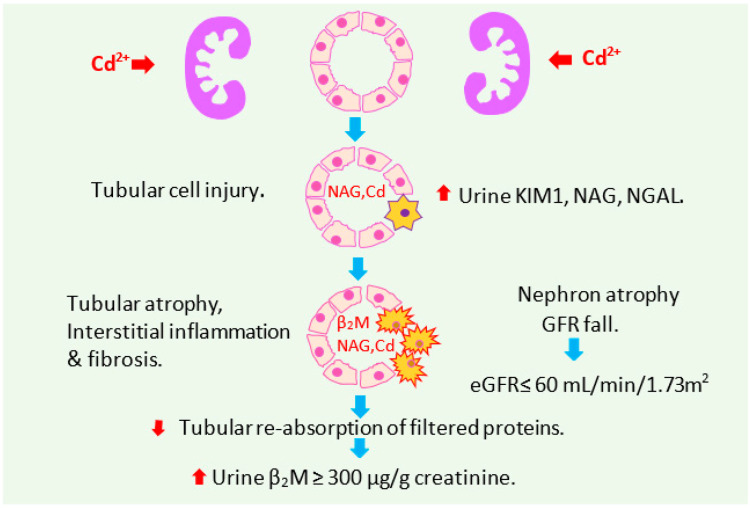
Sequential outcomes of tubular cell toxic injury of cadmium accumulation in kidneys. Cd inflicts tubular cell injury at low intracellular concentrations, and the toxicity intensifies as Cd concentration rises [40]. Tubular injury disables glomerular filtration, causing nephron atrophy, glomerulosclerosis, and interstitial inflammation and fibrosis [40,55,56]. A reduction in tubular reabsorption of filtered proteins, RBP and β_2_MG follows tubular atrophy and a substantial nephron loss. Abbreviation: KIM1, kidney injury molecule 1; NAG, N-acetyl-β-D-glucosaminidase, NGAL, neutrophil gelatinase associated lipocalin; RBP, retinal binding protein; β_2_M, β_2_-microglobulin.

**Table 1 ijerph-19-15697-t001:** Characteristics of the study subjects, stratified by residential location and sex.

Parameters	All Subjects *n* 734	Low Exposure (Bangkok)		High Exposure (Mae Sot)	
		Males, *n* 100	Females, *n* 100	Males, *n* 189	Females, *n* 345
Smoking (%)	42.8	47	0 ***	81.5	32.8 ***
Hypertension (%)	31.7	27	12 **	30.2	39.7 *
Diabetes (%)	1.5	0	0	3.2	1.4
Age, years	48.1 ± 11.0	36.2 ± 10	42.3 ± 9.5 ***	53.3 ± 11.2	50.4 ± 8.2 *
BMI, kg/m^2^	23.2 ± 3.8	23.2 ± 3.5	23.1 ± 3.8	22.0 ± 3.2	23.8 ± 4.1 ***
eGFR ^a^, mL/min/1.73 m^2^	91.0 ± 21.7	104 ± 17	106 ± 14	83 ± 23	87 ± 21 *
Low eGFR ^b^ (%)	9	0	0	15.3	10.7
Plasma creatinine, mg/dL	0.85 (0.74)	0.92 (0.73)	0.67 (0.93) ***	1.04 (1.02)	0.79 (0.90) ***
Urine creatinine, mg/dL	85.59 (1.54)	62.27 (1.68)	48.56 (1.70)	122.50 (1.40)	90.90 (1.50) ***
Urine NAG, U/L	5.48 (1.81)	2.11 (1.93)	1.95 (1.91)	7.83 (1.65)	8.02 (1.81)
Urine β_2_M, µg/L	112.72 (3.10)	4.95 (3.70)	3.52 (3.39)	473.79 (2.88)	340.18 (2.63)
Urine Cd, µg/L	2.34 (2.32)	0.24 (2.83)	0.22 (2.84)	6.34 (2.08)	5.14 (1.92) *
Normalized to E_cr_ as E_x_/E_cr_ ^c^					
E_NAG_/E_cr_, µg/g creatinine	6.40 (1.56)	3.38 (1.73)	4.01 (1.56)	6.39 (1.42)	8.83 (1.57) ***
E_β2M_/E_cr_, U/g creatinine	130.52 (2.97)	7.96 (3.51)	7.26 (3.47)	387.74 (2.48)	374.25 (2.62)
E_Cd_/E_cr_, µg/g creatinine	2.73 (2.09)	0.39 (2.58)	0.46 (2.47)	5.18 (1.83)	5.66 (1.82)
E_NAG_/E_cr_ ≥ 4 U/g creatinine (%)	76.2	48.0	60.0	79.4	87.2 *
E_β2M_/E_cr_ ≥ 300 µg/g creatinine (%)	39.8	7.0	1.0 *	47.6	46.4
Normalized to C_cr_ as E_x_/C_cr_ ^d^					
E_NAG_/C_cr_ × 100, U/L filtrate	5.42 (1.59)	3.12 (1.70)	2.69 (1.71) *	6.64(1.50)	6.98 (1.56)
E_β2M_/C_cr_ × 100, µg/L filtrate	110.5 (15.3)	7.34 (3.48)	4.86 (3.55)	402.82 (2.91)	295.87 (2.63)
E_Cd_/C_cr_ × 100, µg/L filtrate	2.31 (2.15)	0.36 (2.53)	0.31 (2.61)	5.38 (2.01)	4.47 (1.83) *
E_NAG_/C_cr_ × 100 ≥ 4 U/L filtrate (%)	66.2	41.0	21.0 *	81.5	78.3
E_β2M_/C_cr_ × 100 ≥ 300 µg/L filtrate (%)	37.1	7.0	0 **	51.3	48.7

*n*, number of subjects; BMI, body mass index; eGFR, estimated glomerular filtration rate; E_x_, excretion of x; cr, creatinine; C_cr_, creatinine clearance; β_2_M, β_2_-microglobulin; NAG, N-acetyl-β-D-glucosaminidase, Cd, cadmium; ^a^ eGFR, was determined by Chronic Kidney Disease Epidemiology Collaboration equations [33]. ^b^ Low eGFR was defined as eGFR ≤ 60 mL/min/1.73 m^2^. ^c^ E_x_/E_cr_ = [x]_u_/[cr]_u_; ^d^ E_x_/C_cr_ = [x]_u_[cr]_p_/[cr]_u_, where x = Cd, β_2_M or NAG [36]. Data for age, eGFR and BMI are arithmetic means ± standard deviation (SD). Data for all other continuous variables are geometric mean (geometric standard deviation) values. Data for BMI values are from 709 subjects; data for all other variables are from 734 subjects. For each test, *p* ≤ 0.05 identifies statistical significance, determined by Pearson’s Chi-square test for percentage differences and the Mann–Whitney U-test for mean differences between men or women of low- vs. high-exposure locations. * *p* = 0.010–0.035; ** *p* = 0.007; *** *p* < 0.001.

**Table 2 ijerph-19-15697-t002:** Characteristics of the study subjects stratified by eGFR values.

Parameters	eGFR, mL/min/1.73 m^2^			
	Level 1, *n* 410	Level 2, *n* 258	Level 3, *n* 66	*p*
Low-exposure area (%)	40.5	13.2	0	<0.001
Females (%)	59.8	63.2	56.1	0.494
Smoking (%)	34.6	50.4	63.6	<0.001
Diabetes (%)	0	2.7	6.1	<0.001
Hypertension (%)	29.5	32.9	40.1	0.159
Age, years	43.0 ± 9.0	52.2 ± 8.3	63.2 ± 11.7	<0.001
BMI, kg/m^2^	23.5 ± 3.7	23.0 ± 3.9	22.0 ± 4.2	0.020
eGFR ^a^, mL/min/1.73 m^2^	106.8 ± 9.5	77.2 ± 8.0	46.1 ± 11.0	<0.001
Plasma creatinine, mg/dL	0.73 (0.87)	0.95 (0.75)	1.42 (1.34)	<0.001
Urine creatinine, mg/dL	73.60 (1.60)	102.3 (1.46)	108.4 (1.41)	<0.001
Urine NAG, U/L	4.68 (1.85)	6.22 (1.74)	8.97(1.77)	<0.001
Urine β_2_M, µg/L	50.26 (3.14)	165.43 (2.82)	3439.16 (3.63)	<0.001
Urine Cd, µg/L	1.26 (2.43)	4.40 (2.09)	9.39 (2.30)	<0.001
Normalized to E_cr_ as E_x_/E_cr_ ^b^				
E_NAG_/E_cr_, U/g creatinine	6.35 (1.57)	6.08 (1.52)	8.27 (1.68)	0.049
E_β2M_ /E_cr,_ µg/g creatinine	68.24 (2.98)	161.69 (2.73)	3172.49 (3.58)	<0.001
E_Cd_/E_cr_, µg/g creatinine	1.70 (2.18)	4.30 (1.92)	8.66 (2.12)	<0.001
Normalized to C_cr_ as E_x_/C_cr_ ^c^				
E_NAG_/C_cr_ × 100, U/L filtrate	4.61 (1.56)	5.76 (1.50)	11.73 (1.93)	<0.001
E_β2M_ /C_cr_ × 100, µg/L filtrate	49.58 (2.99)	153.15 (2.74)	4497.44 (3.78)	<0.001
E_Cd_/C_cr_ × 100, µg/L filtrate	1.24 (2.20)	4.08 (1.96)	12.28 (2.44)	<0.001

*n*, number of subjects; BMI, body mass index; eGFR, estimated glomerular filtration rate; E_x_, excretion of x; cr, creatinine; C_cr_, creatinine clearance; β_2_M, β_2_-microglobulin; NAG, N-acetyl-β-D-glucosaminidase, Cd, cadmium; ^a^ eGFR, was determined by Chronic Kidney Disease Epidemiology Collaboration (CKD–EPI) equations [33] and eGFR levels 1, 2 and 3 corresponded to eGFR ≥ 90, 61–89, and ≤ 60 mL/min/1.73 m^2^, respectively. ^b^ E_x_/E_cr_ = [x]_u_/[cr]_u_; ^c^ E_x_/C_cr_ = [x]_u_[cr]_p_/[cr]_u_, where x = Cd, β_2_M or NAG [36]. Data for age, eGFR and BMI are arithmetic means ± standard deviation (SD). Data for all other continuous variables are geometric mean (geometric standard deviation) values. Data for BMI are from 709 subjects; data for all other variables are from 734 subjects. For each test, *p* ≤ 0.05 identifies statistical significance, determined by Pearson Chi-Square test for % differences and by Kruskal–Wallis test for mean differences across three eGFR groups.

**Table 3 ijerph-19-15697-t003:** BMDL and BMDU values of cadmium accumulation associated with adverse effects on kidneys according to E_cr_-normalized dataset.

Endpoints	Males			Females		
	BMDL	BMDU	U/L Ratio	BMDL	BMDU	U/L Ratio
Continuous endpoints						
5% Increase of E_NAG/_E_cr_	0.060	0.504	8.44	0.069	0.537	7.79
5% Increase of E_β2MG_/E_cr_	0.019	0.411	22	0.022	0.491	23
5% Decrease of eGFR	1.57 × 10^−6^	0.427	2.7 × 10^5^	1.88 × 10^−5^	0.495	2.6 × 10^4^
Quantal endpoints						
10% Increase in prevalence of tubular cell injury ^a^	1.48 × 10^−3^	0.560	378	3.17 × 10^−4^	0.706	2227
10% Increase in prevalence of defective tubular reabsorption ^b^	0.469	0.973	2.1	0.733	1.29	1.8
10% Increase in prevalence of CKD ^c^	3.260	7.460	2.29	4.980	9.680	1.94

BMDL, and BMDU values of E_Cd_/E_cr_ were as µg/g creatinine. CI, confidence interval; U/L, BMDU/BMDL ratio. U/L ≥ 100 indicated a high degree of statistical uncertainty. ^a^ E_NAG_/E_cr_ ≥ 4 U/g creatinine reflected tubular cell injury. ^b^ E_β2MG_/E_cr_ ≥ 300 µg/g creatinine reflected a defective tubular reabsorption. ^c^ CKD was defined as eGFR ≤ 60 mL/min/1.73 m^2^.

**Table 4 ijerph-19-15697-t004:** BMDL and BMDU values of cadmium excretion levels associated with adverse effects on kidneys according to C_cr_-normalized datasets.

Endpoints	Males			Females		
	BMDL	BMDU	U/L Ratio	BMDL	BMDU	U/L Ratio
Continuous endpoints						
5% Increase of E_NAG/_C_cr_ × 100	0.067	0.394	5.88	0.067	0.399	5.96
5% Increase of E_β2MG_/C_cr_ × 100	0.41 × 10^−4^	0.38	929	0.16 × 10^−3^	0.040	255
5% Decrease of eGFR	0.080	0.714	8.92	0.098	0.812	8.29
Quantal endpoints						
10% Increase in prevalence of tubular cell injury ^a^	0.254	0.458	1.80	0.366	0.614	1.68
10% Increase in prevalence of tubular dysfunction ^b^	0.456	0.807	1.77	0.500	0.805	1.61
10% Increase in prevalence of CKD ^c^	5.060	8.660	1.71	5.150	7.620	1.48

BMDL and BMDU values of E_Cd_/C_cr_ × 100 were in µg/L filtrate. CI, confidence interval; U/L, BMDU/BMDL ratio. U/L ≥ 100 indicated a high degree of statistical uncertainty. ^a^ E_NAG_/C_cr_ × 100 ≥ 4 U/L filtrate reflected tubular cell injury. ^b^ E_β2MG_/C_cr_ × 100 ≥ 300 µg/L filtrate reflected a defective tubular reabsorption. ^c^ CKD was defined as eGFR ≤ 60 mL/min/1.73 m^2^.

## Data Availability

All data are contained within this article.

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
