# Peer review of "The Validity of Benchmark Dose Limit Analysis for Estimating Permissible Accumulation of Cadmium"

_ijerph, 2022, doi:10.3390/ijerph192315697_

Round 1

Reviewer 1 Report

The authors have calculated the benchmark dose limit of cadmium excretion associated with tubulo-glomerular dysfunction.

It is well written. However since the study population includes relatively older populations, it is difficult to generalize the findings to the entire population.

Author Response

Response to Reviewer 1

Comments and Suggestions for Authors

The authors have calculated the benchmark dose limit of cadmium excretion associated with tubulo-glomerular dysfunction.

It is well written. However, since the study population includes relatively older populations, it is difficult to generalize the findings to the entire population

Response:

  • We thank the Reviewer for evaluating our manuscript. We disagree with the Reviewer’s remark questioning the generalizability of findings due to a bias toward older populations.
  • The age distribution histogram below suggests that our findings are generalizable to adult population. This histogram has been inserted as Figure 9S in Supplemental material.
  • We thank the reviewer for raising age consideration issue. Most BMD values for Cd excretion associated with adverse renal effects were derived from those age 50 or older and thus the majority of BMD figures previously published were not generalizable.

Reviewer 2 Report

This is a very statistic-dense report. I find it very difficult to sort through the meaning in all the numbers.  Hopefully my comments & suggestions will be helpful to make these data easier for a broader audience to identify the significance and meaning of the results. 

The abstract starts out saying Cd is highly toxic but is everywhere…perhaps it’s appropriate to clarify that it’s toxic past a certain amount whereas only X amounts are found in the environment?  I'm also concerned about the high variance mentioned and wonder how meaningful data could be extracted...perhaps this just serves as a basis for further studies?  or more explanation needed?

For the intro, I think it would be helpful to state what Cd does biochemically and in particular to the kidney.  How long does it remain in the body? The critical levels are stated - What are considered normal levels of eGFR and Ebeta2M and other standards measured? 

It would be helpful to clearly state the purpose of this study in the last paragraph of the intro.

The tables seem to span pages, can they be placed on the same page?

Section 3.3, Line 222, Should that be Figure 2 instead of Figure 1?

And line 227 should that read Figure 3 not figure 2? 

Might double check all the figure references in the text to be sure.

Figures 2, 3, Raw data. 

The labeling is too small.

If the data are to be shown I would suggest to increase the font size, and then just report the summary data, in particular, just highlight the data you discuss.  For example, do we need to know c, b, b, CEDL, etc?  Perhaps this should all be placed in supplemental, with only a summary graph and summary table. It is too hard to pull out any real meaning from these graphs, and they all look the same.

If you were to test the data based on the level groups, would that reduce the variance?

Fig 2, 3 (e)

Can you explain better in the results section what all the red is?  Is this error bars? I would include explanation of how this is significant to the results reported.

At the end of each section, it would be helpful to have a summary statement that distills out the meaning of all these data.  For example, in section 3.3 remind the reader the importance of measuring Ecd as an indicator of (xx function in kidney) and then say how the data you have relate to that (good or bad?) and the exposure level being high or low.  The range of CI seems really high for most, so it would be good to see what that means to the authors.

Would the CI ranges be smaller if they were divided into the level 1-3 groups?

Are the last 2 references missing or are they just empty lines?

Author Response

Response to Reviewer 2

Comments and Suggestions for Authors

Top comment

This is a very statistic-dense report. I find it very difficult to sort through the meaning in all the numbers.  Hopefully my comments & suggestions will be helpful to make these data easier for a broader audience to identify the significance and meaning of the results. 

Response

  • We thank the Reviewer for evaluation of our manuscript and for constructive comments.
  • We have extensively revised our paper. We have shifted a focus from statistics to public health relevance of BMDL values derived. The title has been changed to read, “The Validity of Benchmark Dose Limit Analysis for Estimating Permissible Accumulation of Cadmium.”

Point 1. The abstract starts out saying Cd is highly toxic but is everywhere…perhaps it’s appropriate to clarify that it’s toxic past a certain amount whereas only X amounts are found in the environment?  I'm also concerned about the high variance mentioned and wonder how meaningful data could be extracted...perhaps this just serves as a basis for further studies?  or more explanation needed?

Response

  • The abstract has been rewritten and it now contains statements regarding the tolerable intake levels and kidney toxicity threshold level of Cd accumulation (lines 14-30)
  • With respect to the high variance, this issue is addressed together with those Points 7 and Point 10.
  • Changes made to the text are in blue.

Point 2. For the intro, I think it would be helpful to state what Cd does biochemically and in particular to the kidney.  How long does it remain in the body? The critical levels are stated - What are considered normal levels of eGFR and Ebeta2M and other standards measured? 

Response: Introduction has been rewritten to contain key elements as suggested.

Point 3. It would be helpful to clearly state the purpose of this study in the last paragraph of the intro.

Response: The purposes of this study have been inserted as suggested. The new title, “The Validity of Benchmark Dose Limit Analysis for Estimating Permissible Accumulation of Cadmium” also states our study objectives.

Point 4. The tables seem to span pages, can they be placed on the same page?

Response

  • In accordance with the journal mandate, tables need to be placed as they are now.

Point 5. Section 3.3, Line 222, Should that be Figure 2 instead of Figure 1? And line 227 should that read Figure 3 not figure 2?  Might double check all the figure references in the text to be sure.

Point 6. Figures 2, 3, Raw data. 

The labeling is too small.

If the data are to be shown I would suggest to increase the font size, and then just report the summary data, in particular, just highlight the data you discuss.  For example, do we need to know c, b, b, CEDL, etc?  Perhaps this should all be placed in supplemental, with only a summary graph and summary table. It is too hard to pull out any real meaning from these graphs, and they all look the same.

Point 7. If you were to test the data based on the level groups, would that reduce the variance?

Point 8. Fig 2, 3 (e)

Can you explain better in the results section what all the red is?  Is this error bars? I would include explanation of how this is significant to the results reported.

Point 9.

At the end of each section, it would be helpful to have a summary statement that distills out the meaning of all these data.  For example, in section 3.3 remind the reader the importance of measuring Ecd as an indicator of (xx function in kidney) and then say how the data you have relate to that (good or bad?) and the exposure level being high or low.  The range of CI seems really high for most, so it would be good to see what that means to the authors.

Response: Points 5, 6, 8 and 9 are pertaining to result presentation, and we address them together here.

  • Data in former Tables 1 and 2 have been reorganized. Former Table 2 becomes Table 1, while former Table 1 becomes Table 2.
  • The BMDL/BMDU values of ECd associated with adverse effects on kidneys have been provided in Tables 3 and 4.
  • Dose-response curves from which the BMDL/BMDU values were extracted to present in Tables 3 and 4 are provided as Supplemental Materials (Figures 1S-8S).
  • Additional paragraphs are inserted to Section 2.5, Benchmark Dose Computation and Benchmark Dose Response (BMR) Setting. The inserts explain multiple dose-response model fittings, rather than a single model.
  • We have explained the utility of the BMDU/BMDL ratio in the reliability evaluation. BMDU/BMDL ratios ≥100 indicate that the BMDL values are unreliable due to a high degree of statistical uncertainty in the dataset.

Point 10. Would the CI ranges be smaller if they were divided into the level 1-3 groups?

Response: Points 7, 10 and part of Point 1 are related. We thus address them together here.

  • A measurement error was unlikely to explain the high variance reported.
  • We attributed the high variance to interindividual differences in muscle mass (the determinant of Ecr), number of surviving nephrons and inflammatory conditions.
  • To eliminate these high variance sources, we normalized excretion of Cd, NAG and β2M to creatinine clearance (Ccr).
  • Ccr-normalization depicts the amount of Cd, NAG and β2M excreted per volume of filtrate, which is presumably related to amount of x excreted per nephron.
  • We have now provided in Appendix A the derivation of an equation to obtain Ccr-normalized excretion. It is possible and convenient to calculate ECd/Ccr Eβ2M/Ccr and ENAG/Ccr without measuring creatinine clearance.
  • We provide in Appendix B an explanation that Ccr-normalization is unaffected by muscle mass. Therefore, it eliminates variation in ratios due to differences in creatinine excretion.  The result is lower BMDU/BMDL ratios and a higher degree of statistical significance (Tables 3 vs. Table 4).
  • In a stratification of subjects by eGFR levels (Table 2), Eβ2M was dramatically increased in those whose functioning nephrons were destroyed, evident from eGFR ≤ 60 mL/min/1.73m2.

Point 11. Are the last 2 references missing or are they just empty lines?

Response:

  • The extra 2 lines were in error.

Round 2

Reviewer 2 Report

Review – v2

I think there is good improvement to Version 2, especially in how the tables are presented. I’m able to follow the study more clearly.  That said, there are still several spots that are difficult to read.  Improving those parts indicated below would greatly improve the read-ability of the article and make it easier for the reader to understand how important it is to look at the individual components of kidney function in different groups of people to be sure if they are in danger of Cd toxicity.

Abstract:

I’m not sure the Abstract needs to be so technical. Perhaps it could be summarized a bit simpler to clearly define the rationale/purpose of this research study and what were the most important findings.  Perhaps just state the conflict you observe, followed by “The purpose of this study is to…”.  “Our results show XYZ.  This suggests that …”, might help lead the reader into the paper better.

Introduction:

I enjoyed reading the revised Introduction -- is much easier to understand – I can easily see the conflict and rationale for the study.  Editing the abstract to be that clear will go a long way.

A few notes in the body of the paper…

Line 17, tolerable intake was set by whom?

Line 21-22, What is the level of reduction and increased excretion that would define the adverse effects or BDML -Cd values?

Line 24, percent of (what) of the smokers?  Or maybe this sentence is just confusing and needs to be more clearly written.

Line 49, grammar issue “was estimated…”; also, is this estimate for a specific group of people, or worldwide?

Line 77, is any increase in Enag an indicator?  Does the Enag value take care of all those 3 injury indicators mentioned?

Line 79, …”versus to”. Do I understand this to be read as “…, versus normalizing those indicators to...”  ?

Line 187-188 reads a bit confusing until I see it in the table…percentages of current smokers… perhaps this is easier to read broken into 2 sentences?  42.8% of the total participants were smokers to include those who had quite less than 10 years. 31.7% of participants had hypertension and only 1.5% had diabetes. 

Line 188, typo, “Th..”

Line 206, might be helpful to remind the reader of how these categories were established (as defined by X guideline) and whether the low or the highest value is the healthiest.

Discussion:

I like the first paragraph in the discussion.

Overall, the discussion appears to be a good summary of the data and shows how it fits with the need for looking at kidney toxicity by Cd differently.

I think it would be good to describe figure 1 in the discussion with a bit more detail, in line with how it relates to the data in this article.

Author Response

Response to Round 1 Review

Top Comment: Review – v2

I think there is good improvement to Version 2, especially in how the tables are presented. I’m able to follow the study more clearly.  That said, there are still several spots that are difficult to read.  Improving those parts indicated below would greatly improve the read-ability of the article and make it easier for the reader to understand how important it is to look at the individual components of kidney function in different groups of people to be sure if they are in danger of Cd toxicity.

Response;

  • We thank the Reviewer for additional comments and suggestions, and for giving us another opportunity to improve our manuscript.

Point 1: Abstract and Introduction

I’m not sure the Abstract needs to be so technical. Perhaps it could be summarized a bit simpler to clearly define the rationale/purpose of this research study and what were the most important findings.  Perhaps just state the conflict you observe, followed by “The purpose of this study is to…”.  “Our results show XYZ.  This suggests that …”, might help lead the reader into the paper better.

Response:

  • In accordance with the Reviewer’s suggestions, parts of the abstract have been rewritten.

I enjoyed reading the revised Introduction -- is much easier to understand – I can easily see the conflict and rationale for the study.  Editing the abstract to be that clear will go a long way.

A few notes in the body of the paper…

Point 1.1: Line 17, tolerable intake was set by whom?

Response

  • The authority responsible for derivation of a tolerable intake of Cd has been provided in the Introduction.

Point 1.2. Line 21-22, What is the level of reduction and increased excretion that would define the adverse effects or BDML -Cd values?

Response: The definition of BMDL has now been provided in the Introduction (lines 73-74) as quoted below.

  • For a continuous endpoint, the lower 95% confidence bound of BMD, termed BMDL value derived at 5% BMR has been viewed as NOAEL equivalent or the level below which adverse effect could be negligible [25].

Point 1.3: Line 24, percent of (what) of the smokers?  Or maybe this sentence is just confusing and needs to be more clearly written.

Response:

  • The referred sentence has been reworded to read as below.
  • “Data were from 289 males and 445 females, mean age of 48.1 years of which 42.8% were smokers, while 9% had chronic kidney disease (CKD), and 31.7% had hypertension.”

Point 1.4: Line 49, grammar issue “was estimated…”; also, is this estimate for a specific group of people, or worldwide?

Response:

  • Grammatical errors have been corrected. The estimate is generalizable to the general human population.

Point 1.5. Line 77, is any increase in ENAG an indicator?  Does the ENAG value take care of all those 3 injury indicators mentioned?

Response:

  • In case of multiple toxicity endpoints, health guidance values for Cd exposure should be derived from the most sensitive one which is the one with the lowest BMDL value.
  • Thus, we have inserted below statement in the text (lines80-82).

“We postulated that the BMDL value of ECd for the ENAG endpoint was the lowest, given that the ENAG represents an early sign of the cytotoxicity of Cd.”

Point 1.6. Line 79, …”versus to”. Do I understand this to be read as “…, versus normalizing those indicators to...”  ?

Response: For clarity, we have rewritten our aim # 3 as quoted below.

  • The third aim was to demonstrate an impact of the methods used to normalize Cd, NAG and β2M excretion rates. Accordingly, we assess the validity of BMDL values of ECd obtained from normalizing excretion rates of Cd, NAG and β2M to creatinine clearance (Ccr) as Ex/Ccr and to the excretion of creatinine (Ecr) as Ex/Ecr, where x = Cd, NAG or β2

Point 3: Results

Point 3.1. Line 187-188 reads a bit confusing until I see it in the table…percentages of current smokers… perhaps this is easier to read broken into 2 sentences?  42.8% of the total participants were smokers to include those who had quite less than 10 years. 31.7% of participants had hypertension and only 1.5% had diabetes.

Response: The corrections have been undertaken as suggested.

Point 3.2: Line 188, typo, “Th..”

Response: A typo error has been corrected.

Point 3.3: Line 206, might be helpful to remind the reader of how these categories were established (as defined by X guideline) and whether the low or the highest value is the healthiest.

Response: CKD staging according to eGFR levels have been provided in the text as quoted below.

  • In current practice, staging of CKD is based on eGFR values, where CKD stages 1, 2, 3a, 3b, 4, and 5 correspond to eGFR of 90–119, 60–89, 45–59, 30−44, 15–29, and < 15 mL/min/1.73 m2, respectively [33]. For dichotomous comparisons, CKD is defined as eGFR ≤ 60 mL/min/1.73 m2 [33].

Point 4: Discussion:

I like the first paragraph in the discussion.

Overall, the discussion appears to be a good summary of the data and shows how it fits with the need for looking at kidney toxicity by Cd differently.

I think it would be good to describe figure 1 in the discussion with a bit more detail, in line with how it relates to the data in this article.

Response:  We have elaborated the hypothesis of Cd-induced nephropathy deduced from BMDL values of ECd for three indicators of renal effects.  One new reference has been added.

Barregard, L.; Sallsten, G.; Lundh, T.; Mölne, J. Low-level exposure to lead, cadmium and mercury, and histopathological findings in kidney biopsies. Environ. Res. 2022, 211, 113119.